

# Hydro-Climatic Modelling of an Ungauged Basin in Kumasi, Ghana

Marian Amoakowaah Osei[1], Leonard Kofitse Amekudzi[1], David Dotse Wemegah[1], Kwasi Preko[1], Emmanuella Serwaa Gyawu[1], and Kwasi Obiri-Danso[1]

[1]Kwame Nkrumah University of Science and Technology, Ghana

*Correspondence to:* Marian A. Osei (marianosans92@gmail.com)

**Abstract.** The 13 km$^2$ Owabi catchment provides about 20% of water needs of the Kumasi metropolis has in recent times been prone to high anthropogenic activities, a source of worry to water resource management. A complementary hydro-climatic study of Owabi watershed has been carried out using Soil-Water-Assessment-Tool (SWAT) with the aim of simulating the stream-flow and water balance of the watershed and to predict its future state. The QGIS interface was used to launch SWAT for QSWAT. Stream-flow output from the model was calibrated against an empirically derived stream-flow dataset for Owabi and the efficacy of the technique tested. The SUFI-2 algorithm was used for calibration and validation on both daily and monthly temporal resolutions. Water loss from the catchment was due to evapotranspiration process followed by surface runoff. The model showed better prediction and low uncertainty for both calibration and validation at the monthly than daily timescale. From 2020 to 2050 under Representative Concentration Pathway 8.5 (RCP8.5), catchment water loss is expected to shift from the dominant evapotranspiraton to surface runoff. This would lead to increases in water yield and stream-flow amount. In general, the use of the SWAT model for hydrological assessment of the Owabi catchment has been successful and further studies on the assessment of water quality and pollution is currently being undertaken to provide a holistic view of water resource management at the catchment. This would aid effective decision making by water resource managers and boost water production for the Kumasi metropolis in the long-term.

## 1 Introduction

Rainfall plays a significant role in the hydrologic cycle and is an essential resource for global socioeconomic activities. Major stakeholders depend on various aspects of the cycle on different time scales. For instance, rain-fed agriculture (dominant in Africa) has been found to operate well if soil moisture is replenished at least every 15 days. Stream-flow which is also important for flood control, hydro-power, navigation and ecological factors has its high and low extremes being controlled by rainfall and groundwater base-flow respectively (Lofgren and Gronewold, 2014).

Water planners and managers rely on assumptions that past and future climatic trends will be the same and hence, water supply systems such as dams are built with these assumptions in mind (Mukheibir, 2007). However, current climate change and variability is offsetting precipitation globally, posing a dire threat to water resource management (WRM). According to



studies by Kankam-Yeboah et al. (2012) [and see references therein], without climate change considerations, Ghana is likely to be water deficient by 2025. This situation is projected to exacerbate with increased anthropogenic activities negatively impacting future water resources by restricting their use to meet growing demand (Kankam-Yeboah et al., 2012).

WRM problems have been tackled by many countries through the Integrated Water Resource Management (IWRM) initiative. IWRM presents a holistic approach in water management by providing a stronger coordination between stakeholders in planning and managing water resources in river basins (Zhang et al., 2014). Ghana's implementation of IWRM at the river basin level begun with the most 'water stressed' basins in the country (Water Resource Commission, 2012). One of such basins is the Pra of which the Owabi catchment serves as a sub-tributary. The dam within the catchment was designed to produce 20% of total potable water requirement of the Kumasi metropolis and its environs. However, increasing human activities including deforestation, sandwinning, illegal logging, indiscriminate dumping of waste has reduced the quality and quantity of water production at the dam. Consequently, this has imposed severe threat on water resource and ecosystem mangement. In order to understand the impacts of human activities on the different hydrological processes, a hydro-climatic modelling assessment was carried out.

The SWAT model incorporates a geographical information system (GIS) interface to give meaningful insights into the water balance, sediments and pollutant transfer in a drainage network (Uzeika et al., 2011). The SWAT model is prominent for its continuous long-term simulations of hydro-climatic variables (Sudjarit et al., 2015) as well as satisfying its developmental aim of testing and predicting water and sediment routing in ungauged basins (Gayathri et al., 2015). It is capable of evaluating the effects of best management practices on water resources in both large and small river basins [see (Uzeika et al., 2011; Sudjarit et al., 2015; Shope et al., 2014; Me et al., 2015)] and has returned favourable performance rate when calibrated.

For instance, Abraham et al. (2007) calibrated and validated the SWAT model for an Ethiopian watershed and found slight under-and over-estimation of peak flows for some years. Nonetheless, the overall performance of the model was good for watershed simulations. Govender and Everson (2005) used the manual calibration technique to model stream-flow for two catchments in South Africa where the increasing demand for timber has drastically changed land use/cover and hydrological processes. Although there was a good response between simulated and observed stream-flows, the model was unable to account for evapotranspiration losses. Schuol and Abbaspour (2006) deduced that the model has a huge potential for freshwater quantification after application to a four million $\mathrm{km}^2$ West African catchment. A study by Begou et al. (2016) at the Bani catchment revealed that calibration at the subbasin scale resulted in better performance than using global parameter set. Furthermore, the model performed well on daily and monthly scales with a good predictive uncertainty, but sometimes highly overestimate potential evapotranspiration.

The aim of this study therefore, was to model the hydro-climate of the Owabi catchment. Specifically, we utilised the SWAT model to simulate stream-flow and establish the water balance for the catchment. The influence of parameter settings on stream-flow was determined through sensitivity/uncertainty analysis; calibration and validation were also performed using SWAT-CUP (SWAT Calibration and Uncertainty Prediction) tool. Finally, we used the SWAT model to forecast the state of hydro-climatic variables at the catchment. The remaining part of this paper is structured as follows; section 2 presents the methodology, results and discussions in section 3 and conclusion given in section 4.



## 2 Methodology

### 2.1 Study Site

The Owabi catchment (Figure 1) has been designated since 1988 as the only inland Ramsar site in Ghana. It comprises of the forest reserve (sanctuary) and the Owabi waterworks and covers about 13 $km^2$ of land area. Its location is between latitudes 6.7292° N and 6.7519° N and longitudes 1.7139° W and 1.6704° W. The forest reserve is a secondary forest enclosing a water reservoir (Commission, 2014). It is one of the smallest conservation sites in Ghana and its protection responsibility lies solely with the Department of Game and Wildlife (Adubofour, 2011). Nonetheless, protection of the site has not deterred high human encroachment and other illegal activities. The hydrological unit is situated in the inner perimeter of the sanctuary (Figure 1). The river was dammed in 1928 with the primary aim of supplying 20% of potable drinking water to the Kumasi metropolis.

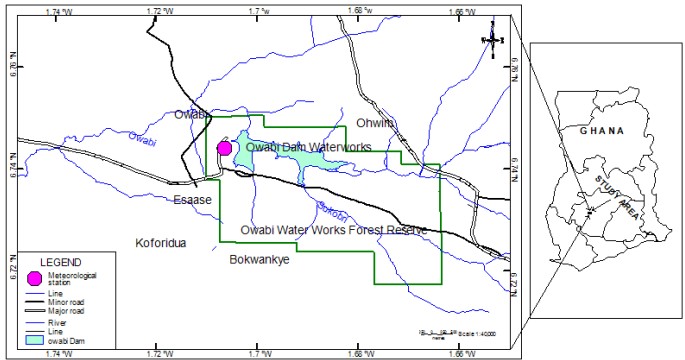

**Figure 1.** The Owabi Catchment area and its environs

The catchment rests in the forest belt of Ghana where rainfall is strongly modulated by the West African Monsoon (WAM) and convective activites resulting from movements of the Intertropical Discontinuity (ITD). The WAM is primarily driven by temperature and energy gradients between the Gulf of Guinea and the Sahara (Amekudzi et al., 2015). Movement of the ITD results in bimodal rainfall regimes in this region with the presence of the river and dense vegetation inducing a microclimate which influences the rainfall patterns, temperature and humidity in the catchment. The mean rainfall is about 1488 mm per annum (Adubofour, 2011) and average monthly temperatures between 24.6 - 27.8° C. Geologically, the area falls within the Birimian meta-sediment of the Kumasi Basin which consists of phyllites, granodiorites, schists, greywackes, tuffs and the associated granitoid (Adubofour, 2011; Commission, 2014) and forest ochrosols are the dominant soil type.

### 2.2 SWAT model description

SWAT is a river-basin scale semi-distributed and physically based model that operates on a daily timescale. It was initially developed by the United States Department of Agriculture to model land management practices on sediment, water and agro-chemical yields in ungauged catchments. The model is highly rated for its computational efficiency and long term continuous




simulations. The important inputs of the model include; rainfall and temperature, digital elevation model, soil and land use maps as well as output simulations of water balance, nutrient and sediment loadings (Arnold et al., 2012b).

The computational hydrology framework of the model is based on the land and water/routing sections of the hydrological cycle. The land division regulates the amount of water, nutrient, sediment and pest loadings into the main channel in every

subbasin. The latter division defines the path of water and sediments through the basin channels to the outlet (Sudjarit et al., 2015). The initial step for watershed simulation is the watershed delineation where subbasins and hydrological response units (HRUs) are defined (Arnold et al., 2012a).

SWAT utilises Equation 1 to simulate the hydrological cycle (Ghoraba, 2015). The hydrological cycle is climate dependent and supplies energy and moisture inputs such as daily rainfall, maximum and minimum air temperatures, wind speed, relative

humidity and solar radiation. The data can be read directly from files by SWAT to produce simulated data at runtime.

$$SW_t = SW_o + \sum_{i+1}^{t}(R_{day} - Q_{surf} - E_a - w_{seep} - Q_{gw})_i \tag{1}$$

where t is time (days), $SW_o$ and $SW_t$ are the initial and final soil water content, $R_{day}$, $Q_{surf}$, $E_a$, $w_{seep}$ and $Q_{gw}$ are the quantities of rainfall, surface runoff, total evapotranspiration and return flow respectively. All parameters have units in mm and $i$ represents the parameter value for a day.

## 2.3   Data Sources

### 2.3.1   GIS Interface and Spatial Dataset

The open source QGIS version 2.6.1 and QSWAT version 1.3 (SWAT2012), downloadable from http://swat.tamu.edu/software/, were employed for the study. Spatial input dataset were obtained from various free internet websites as shown in Table 2. The catchment elevation was within 224 m and 270 m above mean sea level. The input land use map was from MODIS (MODerate

resolution Imaging Spectroradiometer) satellite for a 10 year (2001-2010) regridded global landcover dataset (Broxton et al., 2014) . The soil input has a spatial resolution of 1: 5,000,000 m. All these data were in a raster format and geoprocessed for model input as described in Dile et al. (2016).

Four dominant land-use categories and one soil type were observed within the Owabi catchment as shown in Table 1. The soil type is known as forest ochrosols of the latosol soil group and of the order of climatophytic earths. They are deeply weathered

and consist of thin, dark greyish brown, humus-stained, sandy loam and silt loam topsoils with moderate fine granular structure. The equivalence of this type using the World Reference Base soils is the acrisol/alisols/lixisols (Adjei-Gyapong and Asiamah, 2002).

QSWAT v1.3 applies the TAUDEM V5 (Terrain Analysis Using Digital Elevation Model version 5) to delineate streams within watersheds. A total of thirteen subbasins were delineated and thirty-five HRUs formed based on landuse/soil/slope

thresholds of 20.00/10.00/5.00 [%] respectively (see Figure 2).





**Table 1.** Land-use and soil categories within the Owabi catchment

| Description | QSWAT Code | Percentage within Watershed |
|---|---|---|
| Evergreen Needle-Leaf Forest | FOEN | 75.64 |
| Dryland and Cropland Pasture | CRDY | 18.63 |
| Evergreen Broad-Leaf Forest | FOEB | 1.45 |
| Deciduous Broad-Leaf Forest | FODB | 4.27 |
| Acrisol (soil) | Ao1-ab-1046 | 100 |

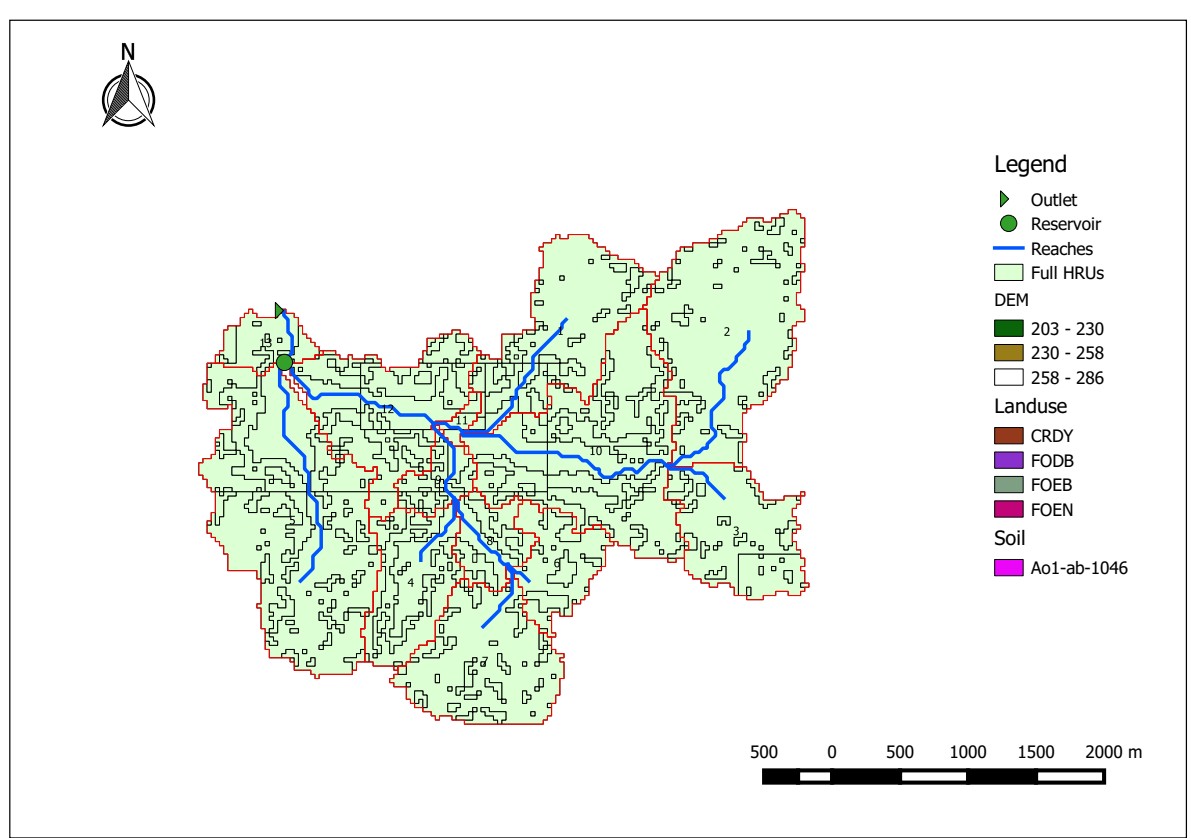

**Figure 2.** Watershed delineation and HRU definition from landuse/soil/slope characteristics for 20/10/5 [%]

## 2.3.2   Meteorological Input

Daily rainfall and temperature (minimum and maximum) were obtained from Ghana Meteorological Agency (GMet) from 1980 to 2015 (Table 2). The data was quality controlled to check for data inhomogeneity as seen in Figure 3. Rainfall gaps





**Table 2.** Input database and sources for the SWAT model

| Data Type | Description | Available Sources |
|---|---|---|
| Digital Elevation Model (DEM) | SRTM 1-Arc-Second Global v3 | USGS |
| Land-Use data | MODIS (15-arc) | USGS-LCI |
| Soil data | Digital global soil map FAOv3.6 | WaterBase |
| Historic climate data (1980 - 2015) | Daily rainfall and temperature | Ghana Meteorological Agency, ARC2, ERA-INTERIM |
| Climate Projection data (2020 - 2050) | Daily rainfall and temperature | CCCMA |

were filled using the African Rainfall Climatological database version 2 (ARC2) at a spatial resolution of 0.1 x 0.1 lat/lon. The choice of ARC2 was due to its narrow error range and longer temporal resolution (Novella and Thiaw, 2013). Temperature gaps were filled using ECMWF's ERA-Interim at a 0.75 x 0.75 lat/lon spatial resolution (Dee et al., 2011). Daily climate projection data (rainfall and temperature) from the Canadian Regional Climate Model at a spatial resolution of 0.44 × 0.44 lat/lon was

5 used for climate forecast. The forecast data were bias corrected before using for SWAT model prediction.

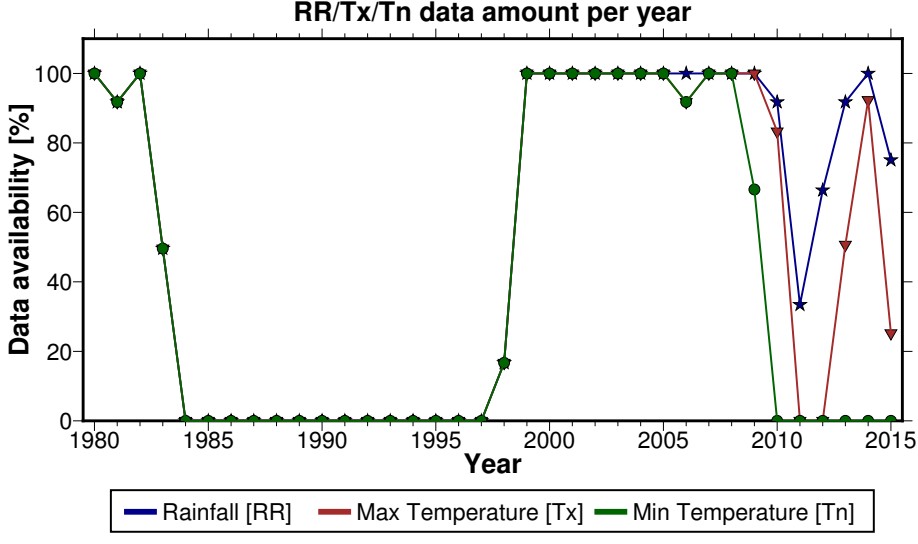

**Figure 3.** Quality assurance of meteorological dataset

Statistical data for the weather generator file were calculated using Wgn Excel Macro v4 downloaded at http://swat.tamu.edu/software/. With the exception of the rainfall and temperature files, two meter dewpoint temperature, 10 meter wind (u) and net solar radiation data were obtained from ECMWF ERA-Interim (Dee et al., 2011) at a 0.125 x 0.125, while the half-hour rainfall was calculated as one-third of the daily maximum rainfall.





### 2.3.3 Hydrological Parameters

The paucity of stream-flow data was a major limitation of this paper for model calibration and validation. In the decade-long review of the prediction in ungauged basins, it has been revealed that regionalisation and other genetic networks can be used for stream-flow determination. However, this paper derived stream-flow estimates from knowledge of the excess rainfall that occurs at the catchment. Firstly, the amount [%] of surface runoff which was approximately 15% was deduced from the 31 year mean monthly water balance of the catchment. This percentage was then used to calculate the daily quantity of surface runoff from rainfall. The runoff percentage was obtained after the initial SWAT model run when the mean monthly water balance was obtained (and available in the output.std file). To obtain the stream-flow, Equation 2 was used. We assumed that this method would work for small watersheds with high slopes and runoff. However, adequate knowledge of the study area is key before employing this technique. We therefore hope that this method would be accurate enough to be used for other small ungauged river basins.

$$Q_e = Q \times A \tag{2}$$

where $Q_e$ represents the stream-flow, $Q$ is the runoff (mm/day) and $A$ is the total basin area. Since stream-flow was derived and not measured, the term 'Estimated' rather than 'Observed' is used throughout the paper.

## 2.4 Calibration, Validation and Uncertainty Assessment

**Table 3.** Statistical Indices and their optimal thresholds for stream-flow according to Moriasi et al. (2007) and Abbaspour (2015)

| Objective function | Threshold (for stream-flow) |
|---|---|
| p-factor | $\geq 0.70$ |
| r-factor | closer to 0 |
| t-stat | larger absolute value |
| p-value | $\leq 0.50$ |
| NSE | $\geq 0.50$ |
| $R^2$ | $\geq 0.50$ |
| PBIAS | $\pm 25\%$ |
| RSR | $\leq 0.70$ |

The autocalibration tool, SWAT-CUP with the Sequential Uncertainty Fitting version 2 (SUFI-2) algorithm was used for calibration techniques. SUFI-2 is known to estimate both parameter and model uncertainties in hydrological models (Abbaspour, 2015). The model was calibrated from 1985 to 1997 and validated from 1998 to 2015 for daily and monthly timescales. Table 3 shows the objective functions used in assessing the overall model performance as well as their required satisfactory thresholds as described in Moriasi et al. (2007) and Abbaspour (2015). The Global sensitivity test for sensitivity analysis was evaluated




using the t-stat and the p-value whiles p and r factors were used for uncertainty analysis at a 95% prediction uncertainty (95PPU). The others included; Nash-Sutcliffe Efficiency (NSE), coefficient of determination ($R^2$), Percentage Bias (PBIAS) and RMSE Standard Deviation Ratio (RSR) [see Appendix A for equation details].

## 3  Results and Discussions

### 3.1  Water Balance

According to Ghoraba (2015) [and references therein], the most important components of the water balance of any hydrological basin are the rainfall, surface runoff, baseflow, lateral flow and evapotranspiration. Unlike rainfall that is easily measureable, the other variables need prediction for their quantification. The SWAT model was run for 35 years, with the first 5 years used as model initialisation. Potential evapotranspiration at the catchment was calculated using the Penman-Montieth technique. The water balance was also visualized using the SWAT-Check tool (Figure 4).

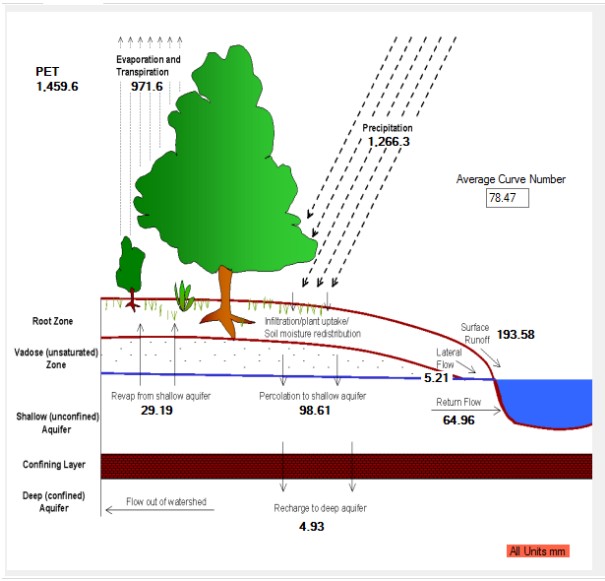

**Figure 4.** A schematic view of the water balance at Owabi Watershed using the SWATCHECK tool

Precipitation distribution was homogeneous within the watershed whiles the expected evapotranspiration (PET) was very high for the period. Water lost at the watershed is regulated by evapotranspiration processes followed by surface runoff. The ratios of actual evapotranspiration (ET) to rainfall and surface runoff to total flow were observed to be 0.77 and 0.73 respectively. Deep recharge and percolation to deep aquifer had no effect on the loss of water and had zero coefficients. Meanwhile, baseflow to total flow was quite minimal (0.21).

The mean seasonal trend of the individual components is observed in Figure 5. ET rates are highest during the northern hemisphere summer monsoon especially July, in response to convective activities associated with the movement of the ITD.





The contrary effect can be observed for the dry months, when the soil moisture is not constantly replenished, causing ET rates to decline. Further analyses showed that the subbasins closest to the outlet experienced decreased ET which is inferred mainly from the prevailing land-use class (CRDY). Footprints showed CRDY to comprise of tropical bamboos and other marshy plants. These plants are characterised by small surface leaf sizes and shapes which directly lowers the amount of transpiration through their stomata. On the other hand, water storage on leaves also reduces leading to less evaporation on plants.

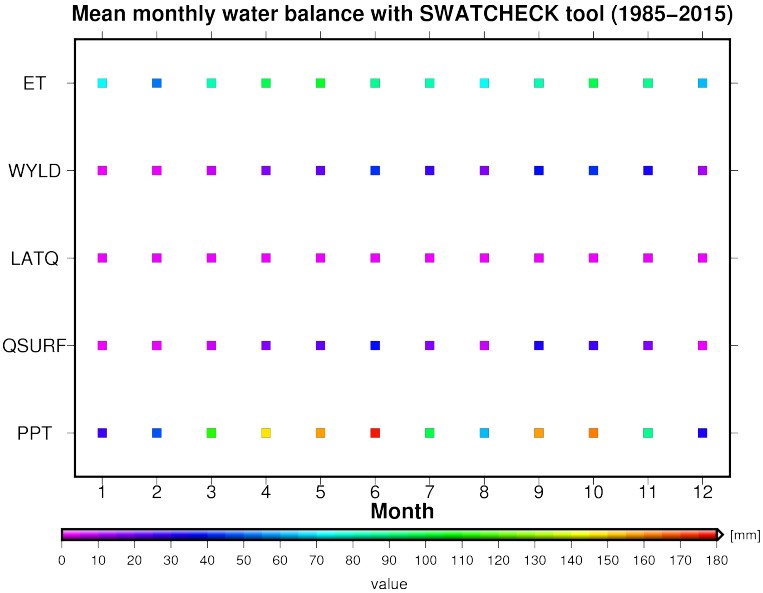

**Figure 5.** Mean monthly water balance from 1985-2015: PPT (Precipitation), QSURF (Surface runoff), LATQ (Lateral flow), WYLD (Water yield), ET (Actual evapotranspiration)

The model simulation showed a bimodal rainfall pattern, the dry season and the little dry spell which is usually observed in August. All trends were in accordance with the rainfall except lateral flow which was observed to be constant regardless of seasonal changes. Forested landscapes are known to have high soil hydraulic conductivity, and this accounts for the low lateral flow observed all year. Water yield (surface runoff + subsurface flow) is an essential component of the cycle and determines how much water leaves the outlet at a given time. It also gives a fair idea on the amount of available water that can be used for socio-economic activities such as water processing and agricultural activities. Again from Figure 5, the amount of water yield is quite low due to the influence of the forest in ET losses. This gives the mode of water processing and utilisation at the catchment by water resource managers (Ghana Water Company Limited).

### 3.2 Calibration, Validation and Uncertainty Results

For calibration analysis, 14 parameters (see table in Appendix 8) that affect surface runoff and baseflow have been selected from literature. It was assumed that all physical processes occuring within the catchment would be accurately captured due to its small size. The global senstivity analysis for daily and monthly calibration is shown in Tables 4 and 5.





GSA on daily timestep showed the most sensitive processes to be the catchment slope (CN2), effective hydraulic conductivity in the main reach (CH_K2) and baseflow alpha factor (ALPHA_BF). On monthly scale three parameters were observed to be highly sensitive: CN2, bulk soil density (SOL_BD) and available soil water content (SOL_AWC). Alternatively, Mannings value for overland flow (OV_N), delay time for groundwater (GW_DELAY) and groundwater revap coefficient (GW_REVAP)

had medium daily sensitivity as well as CH_K2 and saturated hydraulic conductivity (CH_K) for monthly temporal resolution. This information suggests that on all timesteps, water flow is strongly modulated mainly by the catchment slope (CN2) with minimum of 77.0 and maximum of 85.5. These values are concurrent with the rate of high surface runoff which gives a low concentration time of water to the outlet and hence a low capability to minimise peak runoff.

**Table 4.** Daily Senstivity Analysis

| Parameter | t-stat | p-value |
| --- | --- | --- |
| GW_REVAP.gw | 1.65 | 0.10 |
| GW_DELAY.gw | -1.66 | 0.10 |
| OV_N.hru | 1.88 | 0.06 |
| ALPHA_BF.gw | 2.90 | 0.00 |
| CH_K2.rte | -6.64 | 0.00 |
| CN2.mgt | -10.87 | 0.00 |

**Table 5.** Monthly Sensitivity Analysis

| Parameter | t-stat | p-value |
| --- | --- | --- |
| CH_K2.rte | -1.83 | 0.07 |
| SOL_K.sol | 1.86 | 0.06 |
| SOL_AWC.sol | -5.50 | 0.00 |
| SOL_BD.sol | -14.71 | 0.00 |
| CN2.mgt | 15.69 | 0.00 |

### 3.2.1    Stream-flow Analyses on Daily Timescales

From Table 6, the statistical coefficients as well as the 95PPU uncertainty level were quite satisfactory for daily calibration. The p-factor failed to achieve minimum threshold of 0.70 for stream-flow. The 95PPU could only envelope 32% of the estimated stream-flow. High and low peak flows were constantly underestimated as shown through the PBIAS (39.9%). The second 18-year dataset used for validation showed significant improvement in some of the statistics and the simulated stream-flow. Although the width of the 95PPU increased from 8% to 12%, the model was unable to grasp enough of the observations as

evidenced from the p-factor (from 32% to 28%). The bias also decreased its acceptable limit, while $R^2$ and NSE proved the





model to perform quite well. However, low NSE has been reported and linked to the inability of SWAT to capture short, rapid rainfall events that occur in small catchments [(Uzeika et al., 2011) and references therein].

**Table 6.** Results for daily stream-flow calibration and validation

| Objective function | Calibration (1985-1997) | Validation (1998-2015) |
| --- | --- | --- |
| p-factor | 0.32 | 0.28 |
| r-factor | 0.08 | 0.12 |
| $R^2$ | 0.49 | 0.53 |
| NSE | 0.34 | 0.47 |
| RSR | 0.81 | 0.72 |
| PBIAS | 39.9 | 12.4 |
| Mean simulation | 0.03 | 0.06 |
| Mean estimated | 0.05 | 0.07 |

### 3.2.2 Monthly Calibration and Validation

Calibration and validation on monthly timescale (Figure 6) showed a better model performance than daily. During calibration,
the model was able to cover a little over half (54%) of the estimated stream-flow within a wide uncertainty band (1.16). Overall, the estimated stream-flow overpredicts the simulated within a reasonable range as reflected in the PBIAS (9.6%). High and low peak flows were accurately simulated in both wet and dry months in accordance with the rainfall pattern (Figure 7). Low flow periods especially during dry seasons which are controlled by baseflow, groundwater recharge and lateral flow into the Owabi River were adequately represented by the model during calibration. However, groundwater recharge has a low index to
low flow due to the number of days required for the water to reach the river surface and was determined to be approximately 192 days.

The highest peaks during calibration were in September 1987 and September 1988 (0.23 and 0.34 cms), which corresponded with maximum rainfall amounts of 10 and 12 mm per the months (see Figure 7). All the statistical functions showed a satisfactory performance of the model during this temporal calibration.

The monthly validation showed a very good agreement between the simulated and the estimated. This performance conforms with a hydrological review undertaken by Gassman et al. (2007). It is worthy to note that, high peaks were observed to shift from September (calibration) to June (validation). The highest simulated peaks were 0.43 and 0.23 cms (estimated) and obtained in June 2009 with an average rainfall of 17 mm. The objective functions were very satisfactory for the model prediction.

### 3.3 Comparison of uncalibrated with calibrated model

After model optimisation, the new fitted values obtained from the monthly calibration were inserted into the default model and rerun. This was to ensure that the new fitted ranges boosted the model performance for climate and stream-flow prediction.





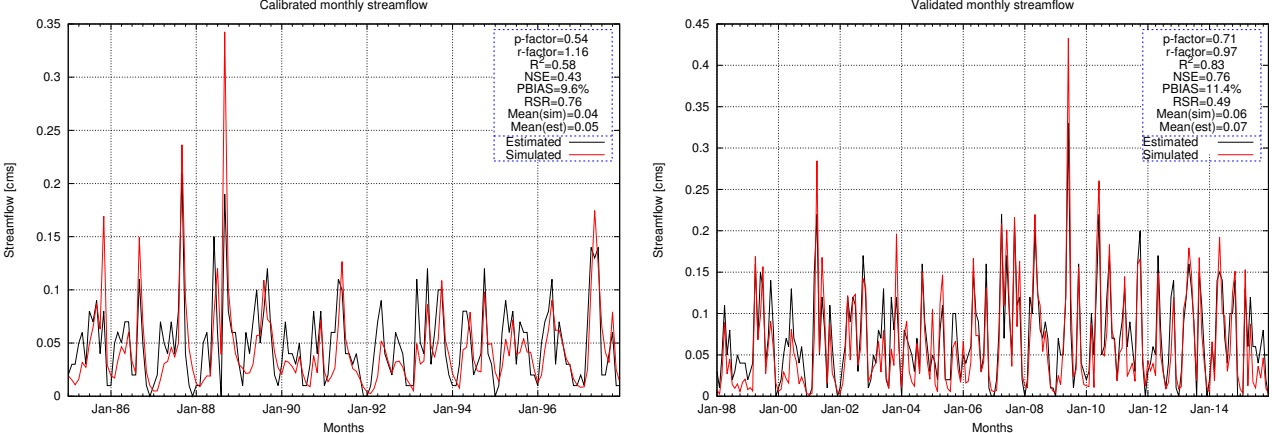

**Figure 6.** Monthly simulated stream-flow for calibration [left: (1985 - 1997)] and the validation period [right: (1998 - 2015)]

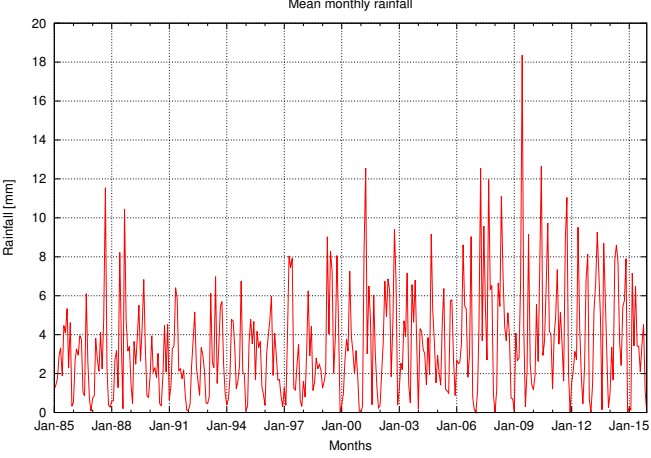

**Figure 7.** A 31-year rainfall time series for the Owabi catchment

The results of the calibrated stream-flow and uncalibrated stream-flow is shown in Figures 8 and 9. The monthly hydrograph of the stream-flow showed a constant underestimation of simulated stream-flow against the estimated.

These deviations were larger for both peak and low flows. The poor representation of the dynamics of the stream-flow at the catchment is further seen in the $R^2$ of 0.27. Meanwhile, the calibrated model showed the contrary, as simulations were found to better predict the observed with a high level of accuracy (R$^2$=0.77). Generally, the pattern of flow was well simulated, but cases existed where the simulation showed high outliers such as in June 2009.





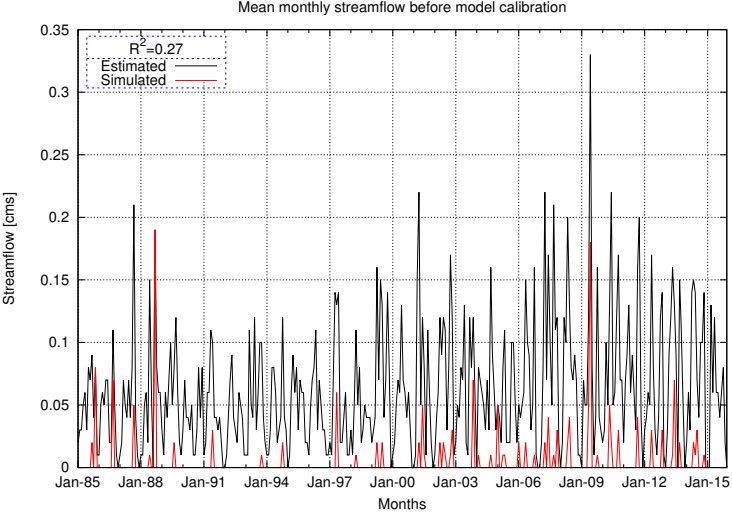

**Figure 8.** Mean monthly uncalibrated stream-flow

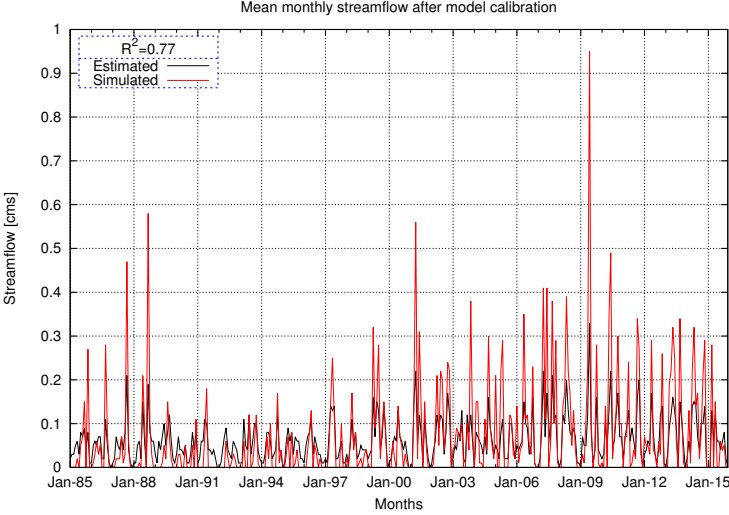

**Figure 9.** Mean monthly calibrated stream-flow

## 3.4   Climate Projections

The Owabi catchment is a very vulnerable site due to the high rate of deforestation and encroachment. Lack of enforcement of

necessary rules by the Ghana forestry commission in protecting the catchment will result in negative implication for the amount

of water available for processing by the Ghana Water Company. As such, we focused our climate change projection on RCP8.5

5  which shows a close representation of socio-economic activites at the catchment. This scenario assumes high population



**Table 7.** Trends of priori and posteriori water balance at the Owabi catchment. The projection parameter is indicated by _p. Deficits are marked in negative(-)

| Months | Rainfall | Rainfall_p | ET | ET_p | LATQ | LATQ_p | SURQ | SURQ_p | WYLD | WYLD_p |
|---|---|---|---|---|---|---|---|---|---|---|
| January | 25.7 | 32.90 | 72.59 | 10.31 | 0.28 | 0.12 | 1.43 | 24.81 | 4.69 | 32.70 |
| February | 48.56 | 15.87 | 50.19 | 15.56 | 0.2 | 0.08 | 1.52 | 4.79 | 2.29 | 11.69 |
| March | 112.98 | 0.70 | 80.58 | 42.61 | 0.27 | 0.06 | 5.71 | 0.00 | 6.34 | 7.08 |
| April | 149.69 | 21.84 | 97.65 | 50.65 | 0.37 | 0.05 | 17.93 | 7.39 | 18.64 | 13.67 |
| May | 158.14 | 105.69 | 103.34 | 50.49 | 0.49 | 0.09 | 22.43 | 44.17 | 24.36 | 50.10 |
| June | 177.84 | 170.27 | 86.74 | 33.51 | 0.56 | 0.17 | 37.99 | 95.35 | 43.15 | 100.57 |
| July | 97.44 | 303.73 | 80.16 | 31.43 | 0.6 | 0.28 | 18.86 | 221.21 | 29.4 | 226.16 |
| August | 62.15 | 247.30 | 71.03 | 32.54 | 0.47 | 0.37 | 5.33 | 179.29 | 15.51 | 184.00 |
| September | 155.79 | 231.07 | 80.44 | 32.89 | 0.44 | 0.42 | 30.16 | 163.18 | 36.78 | 168.28 |
| October | 160.45 | 88.67 | 97.08 | 31.55 | 0.58 | 0.42 | 29.89 | 47.84 | 40.51 | 54.35 |
| November | 85.89 | 16.31 | 88.95 | 21.80 | 0.53 | 0.29 | 19.86 | 6.38 | 33.93 | 13.59 |
| December | 31.26 | 0.31 | 62.43 | 13.04 | 0.41 | 0.19 | 2.46 | 0.04 | 13.18 | 7.83 |
| Mean | 105.49 | 102.89 | 80.93 | 30.53 | 0.43 | 0.21 | 16.13 | 66.20 | 22.39 | 72.50 |
| Mean change | | -2.60 | | -50.40 | | -0.22 | | 50.07 | | 50.10 |

density, relatively slower growth income, moderate technological advancement and improvement in energy intensities (Riahi et al., 2011). This scenario leads to increased energy demand and greenhouse gas emissions in the long-term absence of climate change regulations and policies. RCP8.5 has been classified as the pathway with the highest emissions of greenhouse gases (Riahi et al., 2011).

Table 7 gives the average monthly trends of the historic (1985 - 2015) and the projected (2020 - 2050) water balance for the catchment. Major variations were observed in all the hydrological processes. For instance, maximum mean rainfall during the wet monsoon (May, June, September and October) is expected to change to July, August and September. These projected months are likely to be prone to constant flooding within the catchment especially in July and August. August is known to be a temporary dry spell month during the summer monsoon due to the northward retreat of the ITD. Hence for this month

to experience above than normal rainfall is very alarming. Water loss at the catchment also deviates drastically from actual evapotranspiration to surface runoff. This is expected because of the high deforestation rate that leaves the land surface bare to aid this process. The increase in runoff is also projected to aid water yield increases for the catchment and its peak will follow the rainfall pattern. Hence, if properly harnessed, this could boost water production by the processing plant and for agro-irrigational purposes.

Alternatively, stream-flow (see Figure 10) was projected to increase from a 31 year average of 0.073 cms (priori) to 0.279 cms (posteriori). Although this might be welcoming for water resource management, the likely disadvantage is the reduction in water quality due to high sediment load from increased runoff. This would ultimately stress water production for the Kumasi





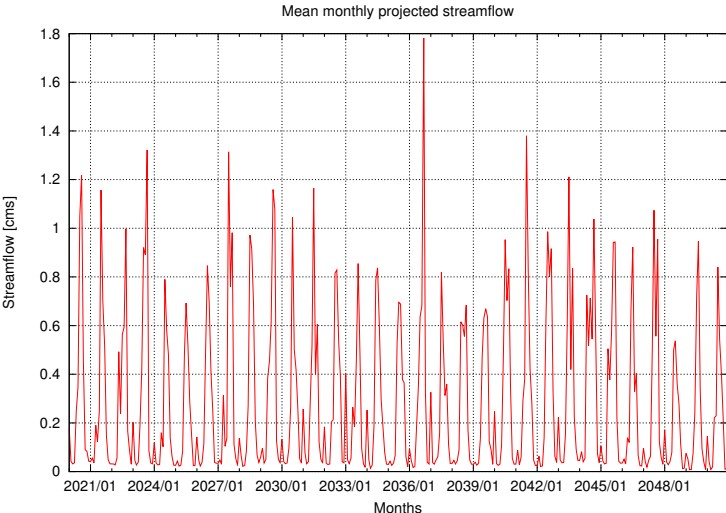

**Figure 10.** Stream-flow forecast using SWAT from 2020 to 2050

metropolis. Hence, this wake-up call should bring together water planning stakeholders for the implementation of appropriate measures to meet future water demands.

## 4 Conclusions

The Soil-Water-Assessment-Tool (SWAT) has been used to model the hydro-climate of the 13 $\text{km}^2$ Owabi catchment. Specifi-
cally, the model simulated historic and projected stream-flow, water balance, as well as model calibration and validation for the catchment. The QGIS interface was used to launch SWAT for QSWAT v1.3. Initial results revealed the forest and topography played major role in water loss at the catchment as evapotranspiration and surface runoff were the most dominant modulating processes. Due to the absence of a hydrometric station at Owabi, a simple empirical technique was used to derive the stream-flow estimates from the surface runoff. The estimates were used for calibration of the model as well as test their efficacy as
alternative stream-flow derivatives for small ungauged catchments.

The SUFI-2 algorithm was used for calibration purposes on both daily and monthly temporal resolutions. The constant parameter found to be most sensitive on both timescales was the catchment slope (CN2) which controlled runoff amounts into the river. The model performed better during monthly calibration than daily. The daily calibration was characterised by small envelop of the estimated stream-flow by the 95PPU, larger biases and quite low correlation and prediction with the simulated
stream-flow. Whereas on the monthly scale, a better model performance ($R^2$ and NSE) was observed for both calibration (0.58 and 0.43) and validation (0.83 and 0.76) at a lower model uncertainty. The biases obtained were also low and within the acceptable stream-flow analysis range as indicated by Moriasi et al. (2007). The use of the estimated stream-flow dataset has shown that on the daily scale, it is characterised by high uncertainty but on monthly scale, biases are compensated for and the




uncertainty reduces. It will be worthwhile to test this empirical method on other small and medium ungauged watershed for

further analyses. However, for this paper, it has proven to be efficient and can be used in addressing water resource management

issues at the catchment.

Lastly, from 2020 to 2050 under RCP8.5, catchment water loss is expected to shift from evapotranspiration to surface
runoff at an unprecedented rate. This would lead to increases in water yield and stream-flow amount. If proper procedures are
implemented, more water can be harvested and stored for both domestic and industrial use. On the other hand, application
of strict regulations can reduce the vulnerability of the catchment to deforestation so as to maintain the current catchment

ecosystem. In general, the use of the SWAT model for hydrological assessment of the Owabi catchment has been successful
and further studies on the assessment of water quality and pollution is currently being undertaken to provide a holistic view
of water resource management at the catchment. This would in the long-term aid effective decision making and boost water
production for the Kumasi metropolis.

## Appendix A: Equations of Objective Functions

$$NSE = 1 - \frac{\sum_i (Q_m - Q_s)_i^2}{\sum_i (Q_{m,i} - \bar{Q_m})^2} \tag{A1}$$

$$R^2 = \frac{\left[\sum_i (Q_{m,i} - \bar{Q_m})(Q_{s,i} - \bar{Q_s})\right]^2}{\sum_i (Q_{m,i} - \bar{Q_m})^2 \sum_i (Q_{s,i} - \bar{Q_s})^2} \tag{A2}$$

$$PBIAS = 100 \times \frac{\sum_{i=1}^n (Q_m - Q_s)_i}{\sum_{i=1}^n Q_{m,i}} \tag{A3}$$

$$RSR = \frac{\sqrt{\sum_{i=1}^n (Q_m - Q_s)_i^2}}{\sqrt{\sum_{i=1}^n (Q_{m,i} - \bar{Q_m})^2}} \tag{A4}$$

where Q is any variable such as discharge, $m$ and $s$ are measured and simulated parameters, bar is the average variable and

$i$ is the $i^{th}$ observed or simulated data.

## Appendix B: Description of parameters

*Author contributions.*  The study was designed by L.K. Amekudzi, D.D. Wemegah and carried out by M.A. Osei. Writeup and data analysis
was done by M.A. Osei and proofreading by all authors.




**Table 8.** Parametric table Arnold et al. (2012a)

| Parameter | Full meaning |
| --- | --- |
| ESCO.hru | Soil evaporation compensation factor |
| SOL_AWC.sol | Available water capacity of the layer of soil |
| CH_N2.rte | Mannings 'n' coefficient |
| SURLAG.bsn | Surface runoff lag coefficient |
| GW_DELAY.gw | Groundwater delay-time |
| GWQMN.gw | Groundwater minimum threshold |
| OV_N.hru | Mannings 'n' value for overland flow |
| GW_REVAP.gw | Groundwater revap coefficient |
| RCHRG_DP.gw | Deep aquifer percolation fraction |
| SOL_BD.sol | Moist bulk density |
| CH_K2.rte | Effective hydraulic conductivity in the main alluvium channel |
| CN2.mgt | Curve number |
| SOL_K.sol | Saturated hydraulic conductivity |
| ALPHA_BF.gw | Baseflow alpha factor |

*Competing interests.* The authors declare no conflict of interest.

*Acknowledgements.* We express our sincere gratitude to Building Stronger Universities Phase 2 (BSU II) and Kwame Nkrumah University of Science and Technology (KNUST) for funding of this study. Mrs. Jamilatou Begou for our training in the use of the SWAT model and Mr. Charles Yorke for making available the climate data from GMet.



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
