# Peer review of "Hydro-Climatic Modelling of an Ungauged Basin in Kumasi, Ghana"

_Hydrology and Earth System Sciences, 2017_

## Referee Comment (RC1) · M. Herrnegger (Referee) · 11 Sep 2017

The manuscript "Hydro-Climatic Modelling of an Ungauged Basin in Kumasi, Ghana" reports on the application of the SWAT model to the Owabi catchment, located about 10 km to the North-West of Kumasi City Centre, Ghana. The Owabi reservoir is part of the study catchment and provides about 1/6 (Erni 2007) to 1/5 (manuscript) of the drinking water demand of the Kumasi metropolis. The authors state that the aim of the study is to simulate the streamflow and water balance of the watershed and to predict its future state. Historical meteorological data is available from 1980 to 2015 for calibration and validation of SWAT. RCP8.5 climate projection data of a single RCM is used to drive the model in the period 2020 to 2050.

Although the manuscript is mostly well structured, I have serious doubts, if it can be published. My general concerns include:

(1) The authors state that the Owabi watershed has an area of 13 km². This number also corresponds to the map shown in Figure 2, when taking the scale bar into consideration (Side note: Legend and scale bar in Figure 1 are too small and not readable). However, Akoto and Abankwa (2014) and Ghana Hydro-Database (2017a) give an area of 60 and 69 km² for the Owabi reservoir catchment, which is about 5 times larger than the value used in the manuscript.

[Figure]

*Figure 1: Owabi reservoir, located about 10 km NW of Kumasi City Centre, including circles with areas of 13 and 65 km²*

Figure 1 shows the Owabi and Barakese reservoirs around Kumasi. The river network is provided by Ghana Hydro-Database (2017b). Unfortunately, catchment polygons are not available and a catchment delineation was not done for this review. As a size comparison the map includes two circles of 13 km² and 65 km². These circles do unfortunately not really shed light on the real catchment area, and from the map it remains unclear if the catchment area used in the manuscript is correct.

There are however indications that the catchment area of 13 km² is not correct, at least if the aim is the modelling of the reservoir catchment: A long-term mean annual streamflow value of 0.073 m³/s is given by the authors (P14L15). This corresponds to a daily water yield of 6 300 m³. This number seems unrealistically too low for two reasons: (i) According to Ghana Hydro-Database (2017a) the reservoir has a storage volume of 2 600 000 m³. With the streamflow given by the authors, it would take over 400 days to fill the reservoir. A reservoir is normally not designed and dimensioned in this way. (ii) According to Erni (2007) the Owabi reservoir provides about 13 500 m³/d for drinking water. This number is over two times larger, compared to the water yield given by the authors. The streamflow in the manuscript would therefore not cover the water demand currently used for the water supply. These considerations ignore issues of residual or environmental flow, which would enhance the quantity of water needed to fill the reservoir or provide water for the water supply.

In summary, a wrong catchment area implies that the results are to be reviewed in a critical way and are likely not showing what they should – the hydrological conditions of the Owabi catchment.

(2) It is clear that data availability is a challenging aspect when modelling areas not only, but also in SSA. In the manuscript, missing discharge data is substituted by, very crudely, multiplying the daily rainfall data with a factor of 0.15, assuming a time constant runoff ratio of 15 %. (Eq. (2) should be something like $Q_{e,t} = P_t * c * w$, with $Q_{e,t}$ [m³/s] - estimated discharge of day t, $P_t$ [mm/d] - rainfall of day t, $c$ - runoff coefficient (0.15) and $w$ - a factor to convert mm/d to m³/s, including catchment area).

The estimated discharge is then used to tune or calibrate the model parameters. This procedure is not legit for several reasons. (i) It completely ignores that runoff ratios change with time (e.g at the beginning of the rainy season, depending on antecedent soil moisture conditions, rainfall intensities or vegetation cover etc., different runoff ratios will be found compared to the end of the rainy season; etc. - reasons why runoff ratios change with time are numerous). (ii) Discharge time series normally show recession curves or falling limbs after peaks that frequently have the

form of an exponential function. They are also continuous in time. Rainfall time series are in contrast discrete. Simply multiplying the rainfall data with a constant factor, especially with daily data, is not an appropriate method to generate an "estimated discharge", since the time series characteristics will be completely different. (iii) The estimated discharge (which is based on the rainfall) is used as comparison to tune model parameters of a model driven by the same input on which the "observed" discharge is based on. This is something like a circular reference and is problematic, to put it kindly. The authors do not, in any way, critically reflect their procedure.

(3) In total, 36 years of historical meteorological data is available, of which 31 years are used for calibration and validation of the simulations (5 years are used as spin-up time). However, the data used is not consistent. About 1/3 of the time series of temperature and rainfall is based on reanalysis data. Judging from Figure 9, the "estimated discharge" seems to be systematically lower in the periods, in which reanalysis data is used (1985-1997/98) compared to periods in which station data is available. This can be a coincidence, but lastly cannot be verified, since the "observed" discharge data is intrinsically based on the (biased?) rainfall.

Additional meteorological parameters (e.g. solar radiation or humidity) are taken completely from reanalysis data. The authors do also not critically discuss this. The additional meteorological data is probably used for estimating potential evapotranspiration (ETp). Why did the authors not use the simpler Hargreaves method also available in SWAT, in which "only" minimum and maximum daily temperature is needed, especially when having the climate projection simulations in mind?

(4) Uncertainties in the simulations are significant. Not necessarily evident in the objective criteria (e.g. Table 6, numbers in Fig. 6) but based on the fact that the discharge estimates used for calibration are not legit (see (2)). The trustworthiness of the model is low. With this model, the future runoff conditions are simulated based on a single climate model projection. Uncertainties in climate projections are, especially concerning precipitation, extremely large. Therefore more than one climate projection should be used as input, simply to get an idea about uncertainties concerning future changes. I also missed a critical discussion by the authors on this topic.

(5) In this context, Table 7 shows the differences between historic and future simulations. Although there is some change in seasonality in rainfall, the annual sums do not differ significantly (1266 mm/a vs. 1234 mm/a). However, actual evapotranspiration (ETa) is reduced by over 62 %, from 671 mm/a to 366 mm/a! This seems unrealistic and the change is

theoretically not reproducible. Since temperatures are expected to increase (e.g. Issahaku et al. 2016), it is likely that the potential ET will be of the same magnitude or, more probable, higher compared to current levels. So the energy available for evapotranspiration will likely increase. It could be that, compared to the past, more months show lower precipitation input, which could lead to lower ETa, since the system becomes water limited. However, this does not seem to be the case, since all months systematically show lower ETa values, independent of the precipitation sums (Table 7). The reason for the lower ETa is insufficiently analysed. The authors state that Penman - Monteith (check spelling in manuscript) was used to estimate ETp. This method is very data intensive and it is unclear, what data (e.g solar radiation, wind speed or humidity) was used for the future simulations.

(6) The Owabi catchment is located near the strongly growing Kumasi metropolis and is therefore exposed to significant human pressures. Changes in the land use and land cover in the catchment is an issue, as stated by the authors but also by Ameyaw and Dapaah (2017) or Forkuo and Frimpong (2012). The latter for example show for the Owabi catchment that the class "Built-up" increased by 26 % and 11 % in the periods 1986-2002 and 2002-2007. At the same time, the class "High Density Forest" was reduced by -23 % and -12 %. These changes took place in the period, for which the simulations were performed. These changes in LULC should be considered, not only for the past, but also for the future projections. (Side note: Why is there no land-use class "water" in Figure 2?)

I will stop my review here, without going into more specific comments.

**References**

Akoto, O., & Abankwa, E. (2014). Evaluation of Owabi Reservoir (Ghana) water quality using factor analysis. Lakes & Reservoirs: Research & Management, 19(3), 174-182.

Ameyaw, Y. & Dapaah G.S. (2017). The Effect of Encroachment on Ecosystem Services Provided By the Owabi Wetland and Wildlife. International Journal of Environmental Sciences & Natural Resources, 4(1): 555628. DOI: 10.19080/IJESNR.2017.04.555628.

Erni, M. 2007. Modelling urban water flows: an insight into current and future water availability and pollution of a fast growing city. Case Study of Kumasi, Ghana. Zürich, Switzerland: Swiss Federal Institute of Technology (ETH). Thesis (MSc), Available from: http://bieb.ruaf.org/ruaf_bieb/upload/3743.pdf. Accessed on 30.08.2017.

Forkuo, E. K., & Frimpong, A. (2012). Analysis of forest cover change detection. International Journal of Remote Sensing Applications, 2(4), 82-92.

Ghana Hydro-Database (2017a). Facts Sheets – Dam & Reservoirs – Ashanti Region Dams. https://www.ghanahydrodata.com/press.php?download_file=DAM-FF-ASR661.pdf. Accessed on 30.08.2017.

Ghana Hydro-Database (2017b). GIS Data – River Basins – Pra River Basin. https://www.ghanahydrodata.com/press.php?download_file=RVB-WGS-PRA377.zip. Accessed on 30.08.2017.

Issahaku, A. R., Campion, B. B., & Edziyie, R. (2016). Rainfall and temperature changes and variability in the Upper East Region of Ghana. Earth and Space Science, 3(8), 284-294.

M. Herrnegger, 11.09.2017

---

## Referee Comment (RC2) · Anonymous Referee #2 · 18 Sep 2017

**Comments: "Hydro-Climatic Modelling of an Ungauged Basin in Kumasi, Ghana"**

**General comments:**

This paper deals with a very important topic on water balance processes and their changes over time according to climate change. This study focuses on the Owabi watershed in Ghana which provides about 20% of drinking water demand of the Kumasi metropolis. The objective of the study is the application of the SWAT model to simulate the streamflow and the water balance of the watershed for the present period (1980-2015) and under future RCP8.5 scenario (2020-2050).

Although the ideas are very interesting and the manuscript well structured, I think this article is not yet mature and still requires work before publication. I have two major concerns:

1) My first general observation concerns the methodology part:

First, in the study site description, the authors should specify that the study site is located in the Sudano-Sahelian region under tropical semi-arid climate. This information allows understanding the systems involved in the region. On the other hand, the watershed is not really indicated in Figure 1. There is just a green rectangular delimitation, which corresponds to the Owabi Water Works Forest Reserve and not to the watershed. When I calculated the study site area, I find a surface, which is about five times larger than the value used in the manuscript (approximately 65 km² based on MNT, SRTM). This value corresponds to what is found in the recent literature (Akoto and Abankwa, 2014) even though actually Frimpong (2011) also gives an area of 13 km². May I feel that this area corresponds to the forest zone? With a clear delimitation of the watershed and a good scale, the question can be solved.

For input data, the authors used MODIS data for the land use map. These data have a wide spatial resolution given the size of the study site. Maybe that higher resolution satellite images (example of Sentinel-2 freely available on the study site) could be used to make a supervised classification and could be compared with MODIS results to take into account the associated errors. In addition, in Figure 2, which represents the watershed, different legend items do not appear on the map. I suggest that authors make several maps to allow the reader to visualize the study site but also to identify the different steps necessary for QSWAT to create the input data. For example: (a) MNT, (b) delineation and discretization of the watershed, (c) soil map, (d) vegetation map and (e) HRUs map. In the same way, a correspondence between soil units and QSWAT look up table is necessary.

Regarding meteorological data, the authors have used ERA-Interim to fill the gaps. Studying the correlation between ERA-Interim data and observed data when available will be interesting to quantify a possible bias. What happened for the gaps before 1989 (start date of ERA-Interim data)? Do you use ERA-40? If this is the case, it will be important to explain because ERA-40 is worse than ERA-Interim and could lead to systematic bias (see for example Mooney et al., 2011). Maybe this could explain the systematically underestimation of surface runoff in Figure 9 on the first simulation years. Please indicate the number of days missing for the different variables and for each year. In any case, I think that the choice to use these data can be justified but must be criticized and uncertainties should be calculated.

I understand the difficulty to obtain discharge data in West Africa but I think that the methodology used in this manuscript is too simplistic. The authors say that "In the decade-long review of the prediction in ungauged basins, it has been revealed that regionalization and other genetic networks can be used for stream-flow determination", but do not give any reference to support this remarks or to detail the

methodology used. If I understand the approach to calculate the "estimated discharge", the discharge was substituted by multiplying the daily precipitation data by a constant runoff coefficient (15%) over time. I think that this methodology is not appropriate for several reason: (a) the runoff coefficient, especially in West Africa, varies over time according to the increase of population and land use change for example but also during the year with the growth of herbaceous; (b) Processes in this region appears at a fine temporal scale (lesser than daily, hourly), so the use of daily precipitation to predict estimated discharge and the use of the same daily precipitation to simulate the runoff with SWAT is not a scientific research approach. I recommend some articles, which develop indirect methods to derive "estimated discharge" over West Africa thanks to reservoirs (see for example Gal et al. 2016, Rodrigues et al., 2013; Sawunyama et al.2006 or Liebe et al., 2005).

2) My second observation is about the novelty of this work and the expected results.

The summary and introduction of this article highlight the increase in human pressure on its environment and the consequences on water budgets. A simulation study (calibration-validation of the model) is then expected, followed by an evaluation of the impact of land use and climate changes on surface runoff. However, the results presented in this manuscript, relate simulation for a single climatic projection (RCP8.5) with the same land cover and soil maps as for the present (if I understand correctly). The conclusions drawn from this study seem to me a little too strong given the methodology used. Testing different climatic and landscape scenarios (hypothetical mixes) will help to consolidate the conclusions of this study as it has been done for the past by Forkuo and Frimpong (2012) for example. The same trend could be used to test a possible land use change for the future.

In page 14, the authors talk about the decrease of evapotranspiration and the increase of surface runoff in the future and gives as explanation that "this is expected because of the high deforestation rate that leaves the land surface bare to aid this process". If the land cover map have been changed, I would agree with the authors' arguments but this is not the case in this study, so I do not understand explanations and results advanced here! If the temperature is expected to increase, evapotranspiration will although increase but ET_p < ET in Table 7. These results remain unclear for me. A scenario section in the methodology part will be useful to explain the approach.

Another point is about the average surface runoff for the present (0.073 $m^3$/s) and for the future periods (0.279 $m^3$/s) given in the manuscript. The Owabi reservoir has an average storage capacity of 2 600 000 $m^3$ (Akoto and Abankwa, 2014; Ghana Hydro-Database, 2017) so 412 days approximately are necessary for filling it in the present against 108 days in the future. According to Maoulidi (2010), the Owabi reservoir provides about 13 600 $m^3$/day for freshwater. This value is larger than the values given by the authors (6 307 $m^3$/day for present). The water supply given by the authors would therefore not cover the water demand.

There are too many questions and approximations in these results. Moreover, there is no explanation, no references to prove or argument the assumptions made and the consequences. I particularly regret the absence of a critical discussion of the results and limitations for the different work's steps.

**Specific and technical comments:**

Figures 1 and 4 are too small and not readable.

Figure 2 should be modified based on previous comments. The delimitation of the reservoir could help the reader.

Figure 5 is not clear and the scale is too larger for some variable. Graphs (with their own legend) for each variable would be better rather than points.

For all Figures, please specify the variable name as it has been done in Figure 5.

Table 1: Please, specify the data resolution.

Table 6: Please, specify the units.

Page 2, Line 11: "management".

Page 3, Line 19: The version of SWAT should be specified. SWAT2012 operates with an hourly, daily, monthly or yearly time step and not only with a daily time step.

Page 4: There is an error in the equation.

Page 6, Line 7: The wind speed is at 10 meters, the authors should note the equation that allowed them to have the wind speed at 2m (variable requested at the input of the model)

Page 8, Line 8: Specify the reason to choose 5 years of warm up and not 10 or 1.

Page 8, Line 9: "Penman-Monteith"

Page 8, Line 11: "Precipitation distribution was homogeneous within the watershed". The explanations are not enough. It is a very important assumption in West Africa that needs to be discussed.

Page 8, Line 13: Is it a runoff by exceeding the soil infiltration capacity (Hortonian runoff)? If yes, please specify in the study site section.

Page 10: Pay attention during the calibration procedure. When modeling future projections, the parameters to be calibrated cannot be variable over time.

Page 10, Line 11: Please, specify the unit.

References: This section lacks many volume numbers, page numbers, editor or doi. Be careful.

Throughout the manuscript, be sure to give references.

**References:**

Akoto, O. and Abankwa, E. (2014), Evaluation of Owabi Reservoir (Ghana) water quality using factor analysis. Lakes Reserv Res Manage, 19: 174–182. doi:10.1111/lre.12066.

Forkuo, E. K., & Frimpong, A. (2012). Analysis of forest cover change detection. International Journal of Remote Sensing Applications, 2(4), 82-92.

Frimpong, A. (2011). Application of Remote Sensing and GIS for Forest Cover Change Detection (A case study of Owabi Catchment in Kumasi, Ghana). An Unpublished M. Sc Thesis Department of Geomatic Engineering, Kwame Nkrumah University of Science and Technology, Kumasi Ghana.

Gal, L., Grippa, M., Hiernaux, P., Peugeot, C., Mougin, E. and Kergoat, L.: Changes in lakes water volume and runoff over ungauged Sahelian watersheds, J. Hydrol., 540, 1176–1188, doi:10.1016/j.jhydrol.2016.07.035, 2016.

Liebe, J., van de Giesen, N. and Andreini, M.: Estimation of small reservoir storage capacities in a semi-arid environment, Phys. Chem. Earth, Parts A/B/C, 30(6–7), 448–454, 2005.

Maolidi, M. 2010. "A Water and Sanitation Needs Assessment for Kumasi, Ghana." MCI Social Sector Working Paper Series, no. 16/2010

Mooney, P. A., Mulligan, F. J., & Fealy, R. (2011). Comparison of ERA-40, ERA-Interim and NCEP/NCAR reanalysis data with observed surface air temperatures over Ireland. International Journal of Climatology, 31(4), 545-557.

Rodrigues, L. N. and Liebe, J.: Small reservoirs depth-area-volume relationships in Savannah Regions of Brazil and Ghana, Water Resour. Irrig. Manag., 2(1), 1–10, 2013.

Sawunyama, T., Senzanje, A. and Mhizha, A.: Estimation of small reservoir storage capacities in Limpopo River Basin using geographical information systems (GIS) and remotely sensed surface areas: Case of Mzingwane catchment, Phys. Chem. Earth, Parts A/B/C, 31(15–16), 935–943, 2006.

---

## Referee Comment (RC3) · Anonymous Referee #3 · 13 Oct 2017

**General comments:**

The authors used the Soil and Water Assessment Tool (SWAT) to simulate streamflow and water balance components for a forested catchment in Ghana. In addition, they use a future climate model to predict changes in streamflow and water balance components, which is the actual aim of the study. I have serious doubts, whether the paper can be published. My two major concerns are:

1. Measured streamflow data were not available. The authors used a very questionable method to derive streamflow. As a result, there is a very high uncertainty in streamflow data that were used for calibration. In general, the methodological approach is often unclear to me. There are many open questions. I will come back to that later in the specific comments. In general, the results are based on many vague and questionable assumptions.

2. The authors used only one climate change scenario, RCP8.5. This scenario is the worst case scenario and I wonder why other more moderate scenarios (RCP6.0, RCP4.5) were not taken into account.

**Specific comments:**

P1, Line 6: I believe, SUFI-2 was only used only for calibration and not for validation as well.

P2, Lines 34-35. This sentence can be deleted

P3, Figure 1: The resolution is too coarse to get all information of the figure. Catchment boundaries would be fine.

P4, line 17: Please, provide the SWAT version number.

P4, line 19-20: what is the grid cell resolution of the land use map?

P4, line 28: what is the grid cell resolution of the DEM?

P4, line 30: This is a small catchment for a SWAT analysis. Soil map has a coarse resolution, land use and DEM I do not know. However, I cannot understand why the authors set thresholds. They lose even more information. They ignore 20% of land use, for example. Please, explain.

P5, Fig 2: This map is unclear too me. There is a green dot representing the dam. The catchment outlet is downstream. That means discharge is completely controlled by dam regulations. The dam itself is approx. 2km from east to west. Am I right that large parts of the delineated catchment is actually open water from the reservoir? So, to avoid confusion and for clarification I suggest to provide different maps. 1. DEM with catchment boundaries, 2. Areal photo or topographic map with REAL river network and delineated catchment boundaries, 3. Land use across the catchment. The legend in the presented map shows DEM and land use but there is NO in formation in the map.

P5: Are meteorological data from the point in fig 1?

P6, l1: Are these data available in daily resolution? Are the local data consistent with the ARC2 data? A figure would be fine, showing prec coming from local and ARC2 data

P6, I3: Are these data available in daily resolution? Are the local data consistent with the ECMWF... data? A figure would be fine, showing temp coming from local and ECMWF data

P6, I6: Which statistical data do the authors mean?

P6, I7: What is the dewpoint temperature for? What about relative humidity?

P7, I1-I14: This is one of the most critical points in this paper. The authors calculated monthly climatic water balance (cwb) from precipitation and potential evapotranspiration. Next, they defined that 15% of monthly cwb is surface runoff. This is weird and calls for explanation! (Also, according to fig6 there is always flow. According to Fig 4, the annual cwb is negative (PET=1459, PREC=1266). That means that there are months with 0mm surface runoff and eventually 0mm total discharge, please explain.). To obtain streamflow, the authors multiply surface runoff with catchment area. Is this an accepted method? Are there references? Please, explain. In addition, the unit surface runoff in eq2 is mm/d. But it is unclear, monthly surface runoff data were converted into daily values. Also, what is the unit of A (basin area)?

P7,I18: There must be two model sets, for daily and for monthly calibration, right? But later on, I see only one parameter set.

P10Table4,5: Are these tables really necessary?

P11, I3: It is unclear to me, why the authors calibrated against daily and monthly streamflow data, and only monthly results are presented. Due to the uncertainty of "measured" streamflow, I would only calibrate at a monthly scale and delete all passages related to daily calibration.

P11, l13: 10 and 12 mm maximum rainfall per month? This is certainly much too low!!!

P12, fig7: maximum monthly rainfall approx. 18mm? This is certainly much too low!!!

P11,I20-21: I do not understand, why fitted parameters were inserted into a default model? "*This was to ensure that the new fitted ranges boosted the model performance for climate and stream-flow prediction.*" I cannot follow, why model performance for climate? Please, explain!

P12,I1: I am confused: There is Fig6 with optimized streamflow and there is fig9 with optimized streamflow data. Predicted streamflow in Fig9 is much higher. Why showing uncalibrated data...

P13,I5: Why did the authors only used RCP8.5 and not others?

Table7: The table captions are not explained. Is it important to show lateral flow? Groundwater flow is not show. Is the high percentage of surface runoff realistic? WYLD in future projection is more then three times larger. Is this realistic? The same for suface runoff....

Figure 10: I believe it is not a good idea to show future projections year by year. Nobody knows, how the wheather will be in 2040. But there are scenarios, how the climate will develop. Therefore, I suggest to compare a 30-years reference period (average) with the future projection (also 30 years, 2021-2050)

---

## Author Comment (AC1) · 18 Oct 2017

The authors would like to thank our reviewer, M. Herrnegger, for his critical evaluation and comments concerning our manuscript titled: *Hydro-climatic modelling of an ungauged basin in Kumasi, Ghana.*

The comments are in normal text and the responses are in bold italic.

**General Comments**

1. The authors state that the Owabi watershed has an area of 13 km². This number also corresponds to the map shown in Figure 2, when taking the scale bar into consideration (Side note: Legend and scale bar in Figure 1 are too small and not readable). However, Akoto and Abankwa (2014) and Ghana Hydro-Database (2017a) give an area of 60 and 69 km² for the Owabi reservoir catchment, which is about 5 times larger than the value used in the manuscript. Figure 1 shows the Owabi and Barakese reservoirs around Kumasi. The river network is provided by Ghana Hydro-Database (2017b). Unfortunately, catchment polygons are not available and a catchment delineation was not done for this review. As a size comparison the map includes two circles of 13 km² and 65 km². These circles do unfortunately not really shed light on the real catchment area, and from the map it remains unclear if the catchment area used in the manuscript is correct. There are however indications that the catchment area of 13 km² is not correct, at least if the aim is the modelling of the reservoir catchment: A long-term mean annual streamflow value of 0.073 m³/s is given by the authors (P14L15). This corresponds to a daily water yield of 6 300 m³. This number seems unrealistically too low for two reasons: (i) According to Ghana HydroDatabase (2017a) the reservoir has a storage volume of 2 600 000 m³. With the streamflow given by the authors, it would take over 400 days to fill the reservoir. A reservoir is normally not designed and dimensioned in this way. (ii) According to Erni (2007) the Owabi reservoir provides about 13 500 m³/d for drinking water. This number is over two times larger, compared to the water yield given by the authors. The streamflow in the manuscript would therefore not cover the water demand currently used for the water supply. These considerations ignore issues of residual or environmental flow, which would enhance the quantity of water needed to fill the reservoir or provide water for the water supply. In summary, a wrong catchment area implies that the results are to be reviewed in a critical way and are likely not showing what they should – the hydrological conditions of the Owabi catchment.

*Response: The Owabi catchment indeed extends to about 69 km² area. The aim of the study was to model the hydrology of the 13 km² forest sub-catchment only. But after careful consideration of your comment, we have decided to incorporate the actual catchment area into the modelling. Hence the entire catchment covering 69 km² is now used. We have also re-run the SWAT ver2012 model and obtained an improved results consistent with values obtained by Erni (2007).*

2. It is clear that data availability is a challenging aspect when modelling areas not only, but also in SSA. In the manuscript, missing discharge data is substituted by, very crudely, multiplying the daily rainfall data with a factor of 0.15, assuming a time

constant runoff ratio of 15 %. (Eq. (2) should be something like $Qe,t = Pt * c * w$, with $Qe,t$[m³/s] - estimated discharge of day t, $Pt$[mm/d] - rainfall of day t, $c$ - runoff coefficient (0.15) and $w$ - a factor to convert mm/d to m³/s, including catchment area). The estimated discharge is then used to tune or calibrate the model parameters. This procedure is not legit for several reasons. (i) It completely ignores that runoff ratios change with time (e.g at the beginning of the rainy season, depending on antecedent soil moisture conditions, rainfall intensities or vegetation cover etc., different runoff ratios will be found compared to the end of the rainy season; etc. - reasons why runoff ratios change with time are numerous). (ii) Discharge time series normally show recession curves or falling limbs after peaks that frequently have the orm of an exponential function. They are also continuous in time. Rainfall time series are in contrast discrete. Simply multiplying the rainfall data with a constant factor, especially with daily data, is not an appropriate method to generate an "estimated discharge", since the time series characteristics will be completely different. (iii) The estimated discharge (which is based on the rainfall) is used as comparison to tune model parameters of a model driven by the same input on which the "observed" discharge is based on. This is something like a circular reference and is problematic, to put it kindly. The authors do not, in any way, critically reflect their procedure.

*Response: The observed streamflow generation method has been removed, as measured streamflow data from a the nearest gauge station (River Offin) is being used. This so as we employ the spatial proximity setting approach aimed at obtaining a more realistic streamflow data for our analysis.*

3. In total, 36 years of historical meteorological data is available, of which 31 years are used for calibration and validation of the simulations (5 years are used as spin-up time). However, the data used is not consistent. About 1/3 of the time series of temperature and rainfall is based on reanalysis data. Judging from Figure 9, the "estimated discharge" seems to be systematically lower in the periods, in which reanalysis data is used (1985-1997/98) compared to periods in which station data is available. This can be a coincidence, but lastly cannot be verified, since the "observed" discharge data is intrinsically based on the (biased?) rainfall. Additional meteorological parameters (e.g. solar radiation or humidity) are taken completely from reanalysis data. The authors do also not critically discuss this. The additional meteorological data is probably used for estimating potential evapotranspiration (ETp). Why did the authors not use the simpler Hargreaves method also available in SWAT, in which "only" minimum and maximum daily temperature is needed, especially when having the climate projection simulations in mind?

*Response: Previously, the Penman-Monteith method was used, which was based on rainfall, temperature, solar radiation and wind. However, the Hargreaves method is currently being used which requires only daily rainfall and temperature (maximum and minimum) for the new model run in both baseline and future projections. Observed climatological rainfall data developed by the Meteorology and Climate Science Unit of KNUST and reported in Aryee et al., (2017) is being used for the model. Although comparison of daily minimum and maximum temperatures (2000 – 2004) from the Owabi station and ECMWF data showed a*

*consistent agreement (R = 0.6), currently, we have used the nearest weather station data from Kumasi Airport to fill in the missing temperature gaps.*

4.  Uncertainties in the simulations are significant. Not necessarily evident in the objective criteria (e.g. Table 6, numbers in Fig. 6) but based on the fact that the discharge estimates used for calibration are not legit (see (2)). The trustworthiness of the model is low. With this model, the future runoff conditions are simulated based on a single climate model projection. Uncertainties in climate projections are, especially concerning precipitation, extremely large. Therefore more than one climate projection should be used as input, simply to get an idea about uncertainties concerning future changes. I also missed a critical discussion by the authors on this topic.

***Response: The high pace of urbanisation and deforestation at the catchment led to the choice of the RCP8.5 as the projection scenario. Notwithstanding, we are also including trends in RCPs 2.6 and 4.5 in the revised manuscript. Uncertainties associated with the simulations will be reviewed and appropriately updated in the revised manuscript.***

5.  In this context, Table 7 shows the differences between historic and future simulations. Although there is some change in seasonality in rainfall, the annual sums do not differ significantly (1266 mm/a vs. 1234 mm/a). However, actual evapotranspiration (ETa) is reduced by over 62 %, from 671 mm/a to 366 mm/a! This seems unrealistic and the change is theoretically not reproducible. Since temperatures are expected to increase (e.g. Issahaku et al. 2016), it is likely that the potential ET will be of the same magnitude or, more probable, higher compared to current levels. So the energy available for evapotranspiration will likely increase. It could be that, compared to the past, more months show lower precipitation input, which could lead to lower ETa, since the system becomes water limited. However, this does not seem to be the case, since all months systematically show lower ETa values, independent of the precipitation sums (Table 7). The reason for the lower ETa is insufficiently analysed. The authors state that Penman - Monteith (check spelling in manuscript) was used to estimate ETp. This method is very data intensive and it is unclear, what data (e.g solar radiation, wind speed or humidity) was used for the future simulations.

***Response: The spelling of Penman-Monteith has been modified in the manuscript. Due to the absence of solar radiation, wind and humidity data at the catchment, 10 meter wind (u) and net solar radiation data were obtained from ECMWF ERA-Interim. This led to the choice of the historic evapotranspiration estimation method to be the Penman-Monteith. For the future projections, only daily rainfall and maximum and minimum temperatures were used. However, in the modified script, the Hargreaves method is being used to ensure that both baseline and projection data are of the same type.***

6. The Owabi catchment is located near the strongly growing Kumasi metropolis and is therefore exposed to significant human pressures. Changes in the land use and land cover in the catchment is an issue, as stated by the authors but also by Ameyaw and Dapaah (2017) or Forkuo and Frimpong (2012). The latter for example show for the Owabi catchment that the class "Built-up" increased by 26 % and 11 % in the periods 1986-2002 and 2002-2007. At the same time, the class "High Density Forest" was reduced by -23 % and -12 %. These changes took place in the period, for which the simulations were performed. These changes in LULC should be considered, not only for the past, but also for the future projections. (Side note: Why is there no land-use class "water" in Figure 2?)

***Response: Different landuse scenarios will be developed for the future projections at the watershed. A new landuse data from the European Space Agency, which characterises the landuse change between 1992 to 2015 at 300 m spatial resolution, has been found be more suitable for the work. This is currently being employed in the study. Landuse category "water" is also presented in the modified manuscript.***

REFERENCES

Aryee, J.N.A., Amekudzi, L.K., Quansah, E., Klutse, N.A.B., Atiah, W.A. and Yorke, C., 2017. Development of high spatial resolution rainfall data for Ghana. *International Journal of Climatology*.

---

## Author Comment (AC2) · 18 Oct 2017

The authors would like to thank our anonymous reviewer for his critical evaluation of the manuscript and comments. They have been very insightful.

**General Comments**

1. My first general observation concerns the methodology part:
   First, in the study site description, the authors should specify that the study site is located in the Sudano-Sahelian region under tropical semi-arid climate. This information allows understanding the systems involved in the region. On the other hand, the watershed is not really indicated in Figure 1. There is just a green rectangular delimitation, which corresponds to the Owabi Water Works Forest Reserve and not to the watershed. When I calculated the study site area, I find a surface, which is about five times larger than the value used in the manuscript (approximately 65 km² based on MNT, SRTM). This value corresponds to what is found in the recent literature (Akoto and Abankwa, 2014) even though actually Frimpong (2011) also gives an area of 13 km². May I feel that this area corresponds to the forest zone? With a clear delimitation of the watershed and a good scale, the question can be solved. For input data, the authors used MODIS data for the land use map. These data have a wide spatial resolution given the size of the study site. Maybe that higher resolution satellite images (example of Sentinel-2 freely available on the study site) could be used to make a supervised classification and could be compared with MODIS results to take into account the associated errors. In addition, in Figure 2, which represents the watershed, different legend items do not appear on the map. I suggest that authors make several maps to allow the reader to visualize the study site but also to identify the different steps necessary for QSWAT to create the input data. For example: (a) MNT, (b) delineation and discretization of the watershed, (c) soil map, (d) vegetation map and (e) HRUs map. In the same way, a correspondence between soil units and QSWAT look up table is necessary. Regarding meteorological data, the authors have used ERA-Interim to fill the gaps. Studying the correlation between ERA-Interim data and observed data when available will be interesting to quantify a possible bias. What happened for the gaps before 1989 (start date of ERA-Interim data)? Do you use ERA-40? If this is the case, it will be important to explain because ERA-40 is worse than ERA-Interim and could lead to systematic bias (see for example Mooney et al., 2011). Maybe this could explain the systematically underestimation of surface runoff in Figure 9 on the first simulation years. Please indicate the number of days missing for the different variables and for each year. In any case, I think that the choice to use these data can be justified but must be criticized and uncertainties should be calculated. I understand the difficulty to obtain discharge data in West Africa but I think that the methodology used in this manuscript is too simplistic. The authors say that "In the decade-long review of the prediction in ungauged basins, it has been revealed that regionalization and other genetic networks can be used for stream-flow determination", but do not give any reference to support this remarks or to detail the methodology used. If I understand the approach to calculate the "estimated discharge", the discharge was substituted by multiplying the daily precipitation data by a constant runoff coefficient (15%) over time. I think that this methodology is not appropriate for several reason: (a) the runoff coefficient, especially in West Africa, varies over time according to the increase of population and land use change for

example but also during the year with the growth of herbaceous; (b) Processes in this region appears at a fine temporal scale (lesser than daily, hourly), so the use of daily precipitation to predict estimated discharge and the use of the same daily precipitation to simulate the runoff with SWAT is not a scientific research approach. I recommend some articles, which develop indirect methods to derive "estimated discharge" over West Africa thanks to reservoirs (see for example Gal et al. 2016, Rodrigues et al., 2013; Sawunyama et al.2006 or Liebe et al., 2005).

*Response: The Owabi catchment indeed extends to about 69 km$^2$ area. The aim of the study was to model the hydrology of the 13 km$^2$ forest sub-catchment only. But after careful consideration of your comments, we have decided to incorporate the larger catchment area into the modelling. The input maps for delineation of watershed, soil, vegetation and HRU maps will be added in the revised manuscript. ERA-INTERIM reanalysis data was used and the timeseries begins from 1979. Although comparison of daily minimum and maximum temperatures (2000 – 2004) from the Owabi station and ECMWF data showed a consistent agreement (R = 0.6), currently, we have used the nearest weather station data from Kumasi Airport to fill in the missing temperature gaps. The correlation analysis between ERA-INTERIM and the observed data can only be performed for the temperature values since there exists no station data for solar radiation, wind and dewpoint temperature. Hence the Hargreaves model has been considered for estimation of evapotranspiration. The number of missing days of data would be incorporated in the revised manuscript. A considerable amount of effort is being put into obtaining discharge data from Owabi, which has not been very fruitful yet. Therefore, applying the spatial proximity setting, streamflow from the nearest gauging station (River Offin) is being used to run the model for the Owabi catchment. A new landuse data from the European Space Agency, which characterises the landuse change between 1992 to 2015 at 300 m spatial resolution, has been found be more suitable for the work.*

2. My second observation is about the novelty of this work and the expected results. The summary and introduction of this article highlight the increase in human pressure on its environment and the consequences on water budgets. A simulation study (calibration-validation of the model) is then expected, followed by an evaluation of the impact of land use and climate changes on surface runoff. However, the results presented in this manuscript, relate simulation for a single climatic projection (RCP8.5) with the same land cover and soil maps as for the present (if I understand correctly). The conclusions drawn from this study seem to me a little too strong given the methodology used. Testing different climatic and landscape scenarios (hypothetical mixes) will help to consolidate the conclusions of this study as it has been done for the past by Forkuo and Frimpong (2012) for example. The same trend could be used to test a possible land use change for the future. In page 14, the authors talk about the decrease of evapotranspiration and the increase of surface runoff in the future and gives as explanation that "this is expected because of the high deforestation rate that leaves the land surface bare to aid this process". If the land cover map have been changed, I would agree with the authors' arguments but this is not the case in this study, so I do not understand explanations and results advanced here! If the temperature is expected to increase, evapotranspiration will although increase but ET_p < ET in Table 7.

These results remain unclear for me. A scenario section in the methodology part will be useful to explain the approach. Another point is about the average surface runoff for the present (0.073 m$^3$/s) and for the future periods (0.279 m$^3$/s) given in the manuscript. The Owabi reservoir has an average storage capacity of 2 600 000 m3(Akoto and Abankwa, 2014; Ghana Hydro-Database, 2017) so 412 days approximately are necessary for filling it in the present against 108 days in the future. According to Maoulidi (2010), the Owabi reservoir provides about 13 600 m3/day for freshwater. This value is larger than the values given by the authors (6 307 m3/day for present). The water supply given by the authors would therefore not cover the water demand. There are too many questions and approximations in these results. Moreover, there is no explanation, no references to prove or argument the assumptions made and the consequences. I particularly regret the absence of a critical discussion of the results and limitations for the different work's steps.

*Response: The high pace of urbanisation and deforestation at the catchment led to the choice of the RCP8.5 as the projection scenario. Notwithstanding, we are also including trends in RCPs 2.6 and 4.5 in the revised manuscript. Previously, the landuse input map was used for the projection of future hydro-climatic trends. However, different landuse scenarios are being considered in hand with the different RCPs to give consolidated conclusions to this study.*

**Specific and technical comments:**

Figures 1 and 4 are too small and not readable.

*Response: Changes have been made in the manuscript and these figures are enlarged.*

Figure 2 should be modified based on previous comments. The delimitation of the reservoir could help the reader.

*Response: Duly noted and these would be shown in the revised manuscript.*

Figure 5 is not clear and the scale is too larger for some variable. Graphs (with their own legend) for each variable would be better rather than points.

*Response: This will be improved.*

For all Figures, please specify the variable name as it has been done in Figure 5.

*Response: This would be done in the revised manuscript.*

Table 1: Please, specify the data resolution.

*Response: The data resolution will be added in Table 2.*

Table 6: Please, specify the units.

*Response: The units will be added.*

Page 2, Line 11: "management".

*Response: The word "mangement" has been modified in the text to "management"*

Page 3, Line 19: The version of SWAT should be specified. SWAT2012 operates with an hourly, daily, monthly or yearly time step and not only with a daily time step.

*Response: The text has been updated.*

Page 4: There is an error in the equation.

*Response: The term "i+1" has been modified to the correct form "i=1"*

Page 6, Line 7: The wind speed is at 10 meters, the authors should note the equation that allowed them to have the wind speed at 2m (variable requested at the input of the model).

*Response: The much simpler Hargreaves ET method is currently being employed and hence the wind variable would not be used anymore.*

Page 8, Line 8: Specify the reason to choose 5 years of warm up and not 10 or 1.

*Response: The 5 years was chosen to ensure uniformity in the baseline and future projections timeseries (both of which have 31 years).*

Page 8, Line 9: "Penman-Monteith"

*Response: The name has been modified.*

Page 8, Line 11: "Precipitation distribution was homogeneous within the watershed". The explanations are not enough. It is a very important assumption in West Africa that needs to be discussed.

*Response: West African rainfall is precarious and has an inhomogeneous distribution. However, based on the size of the study, there was very little variations within the catchment area. Gridded data over Ghana is now available as reported in Aryee et al., (2017). This would be therefore used for the model run over the new catchment area (69 km$^2$).*

Page 8, Line 13: Is it a runoff by exceeding the soil infiltration capacity (Hortonian runoff)? If yes, please specify in the study site section.

*Response: Soil within the study area falls under "D" in the Hydrological Soil Group. This type has very slow infiltration rates when completely wetted. Hence runoff at the catchment is not of the Hortonian.*

Page 10: Pay attention during the calibration procedure. When modeling future projections, the parameters to be calibrated cannot be variable over time.

*Response: This is well noted.*

Page 10, Line 11: Please, specify the unit.

*Response: According to Abbaspour (2015), the p-factor threshold for streamflow is a dimensionless quantity and can be expressed in decimal fraction or percentage (ie. 0.70 or 70%).*

References: This section lacks many volume numbers, page numbers, editor or doi. Be careful. Throughout the manuscript, be sure to give reference**s.**

*Response: The references is being updated accordingly.*

REFERENCES:

Abbaspour, K.: SWAT-CUP, eawag, 2015**.**

Aryee, J.N.A., Amekudzi, L.K., Quansah, E., Klutse, N.A.B., Atiah, W.A. and Yorke, C., 2017. Development of high spatial resolution rainfall data for Ghana. *International Journal of Climatology*.

---

## Author Comment (AC3) · 18 Oct 2017

The authors would like to thank our anonymous reviewer for his critical evaluation of the manuscript and comments. They have been very insightful.

**General Overview**

1. Measured streamflow data were not available. The authors used a very questionable method to derive streamflow. As a result, there is a very high uncertainty in streamflow data that were used for calibration. In general, the methodological approach is often unclear to me. There are many open questions. I will come back to that later in the specific comments. In general, the results are based on many vague and questionable assumptions.

*Response: The observed streamflow generation method has been removed, as measured streamflow data from a the nearest gauge station (River Offin) is being used. This so as we employ the spatial proximity setting approach aimed at obtaining a more realistic streamflow data for our analysis.*

2. The authors used only one climate change scenario, RCP8.5. This scenario is the worst case scenario and I wonder why other more moderate scenarios (RCP6.0, RCP4.5) were not taken into account.

*Response: The high pace of urbanisation and deforestation at the catchment led to the choice of the RCP8.5 as the projection scenario. Notwithstanding, we are also including trends in RCPs 2.6 and 4.5 in the revised manuscript. Previously, the landuse input map was used for the projection of future hydro-climatic trends. However, different landuse scenarios are being considered in hand with the different RCPs to give consolidated conclusions to this study.*

**Specific comments:**

P1, Line 6: I believe, SUFI-2 was only used only for calibration and not for validation as well.

*Response: SUFI-2 was used for both calibration and validation as stated in the text, "The SUFI-2 algorithm was used for calibration and validation on both daily and monthly temporal resolutions."*

P2, Lines 34-35. This sentence can be deleted

*Response: We believe the statement in lines 34-35 are significant, since most manuscripts outline the structure of their study.*

P3, Figure 1: The resolution is too coarse to get all information of the figure. Catchment boundaries would be fine.

*Response: Figure 1 will be modified and the resolution also improved.*

P4, line 17: Please, provide the SWAT version number.

*Response: SWAT version 2012 (SWAT2012). This is updated in the revised manuscript.*

P4, line 19-20: what is the grid cell resolution of the land use map?

*Response: The grid cell resolution for land use map was 500 m x 500 m. However, in the current update of the model, we are using ESA annual 1992-2015 landuse map at a resolution of 300 m x 300 m. This is to account for land use changes occurring within the time period of the study, as well as incorporate the water body (WATR) land use category which was previously not captured by the SWAT model.*

P4, line 28: what is the grid cell resolution of the DEM?

*Response: DEM resolution is 30 m x 30 m. This is updated in the revised manuscript.*

P4, line 30: This is a small catchment for a SWAT analysis. Soil map has a coarse resolution, land use and DEM I do not know. However, I cannot understand why the authors set thresholds. They lose even more information. They ignore 20% of land use, for example. Please, explain.

*Response: The study focussed solely on modelling the forested part of the Owabi catchment. Landuse within the forest remains fairly uniform with no changes within the 13 km² forest cover. However in the updated manuscript, we have taken into consideration the entire catchment area of about 69 km². Therefore to include all land uses, the threshold has been set at a value of 1 %.*

P5, Fig 2: This map is unclear too me. There is a green dot representing the dam. The catchment outlet is downstream. That means discharge is completely controlled by dam regulations. The dam itself is approx. 2km from east to west. Am I right that large parts of the delineated catchment is actually open water from the reservoir? So, to avoid confusion and for clarification I suggest to provide different maps. 1. DEM with catchment boundaries, 2. Areal photo or topographic map with REAL river network and delineated catchment boundaries, 3. Land use across the catchment. The legend in the presented map shows DEM and land use but there is NO in formation in the map.

*Response: Although the outlet is downstream, the dam has a free overflow spillage system. Dam information has been updated in the model run. The paper focussed on the forest hydrology of the catchment, but update is being made to expand the area to cover the actual 69 km² of the catchment. The individual input maps for delineation of watershed, soil, vegetation and HRU maps will be added in the revised manuscript.*

P5: Are meteorological data from the point in fig 1?

*Response: Daily rainfall and temperature records are point data from the study area in Fig 1*

P6, l1: Are these data available in daily resolution? Are the local data consistent with the ARC2 data? A figure would be fine, showing prec coming from local and ARC2 data

*Response: All data were in daily resolution. Both station rainfall data and ARC2 rainfall datasets on the other hand, have been compared for the periods of 2000 – 2004 and there existed quite good agreement (R=0.4). However, observed gridded rainfall data from the station as seen in Aryee et al., 2017 is being used for the model. A figure would be provided when necessary in the revised manuscript.*

P6, l3: Are these data available in daily resolution? Are the local data consistent with the ECMWF data? A figure would be fine, showing temp coming from local and ECMWF data

*Response: All data were in daily resolution. Although comparison of daily minimum and maximum temperatures (2000 – 2004) from the Owabi station and ECMWF data showed a consistent agreement (R = 0.6), currently, we have used the nearest weather station data from Kumasi Airport to fill in the missing temperature gaps.*

P6, l6: Which statistical data do the authors mean?

*Response: These are the daily data needed for generating the weather generator file (an input for the SWAT model) for the study area. The data include; rainfall, maximum and minimum temperatures, solar radiation, dewpoint temperature and wind speed. Statistics such as monthly averages, standard deviations, skew coefficient, among others were calculated from the listed datasets as described in the SWAT2012 input/output documentation manual (Arnold et al., 2012).*

P6, l7: What is the dewpoint temperature for? What about relative humidity?

*Response: Dewpoint temperature was used for the calculations in the weather generator file (WGN). The statistics within the WGN file was then used for simulation of relative humidity for the catchment.*

P7, l1-l14: This is one of the most critical points in this paper. The authors calculated monthly climatic water balance (cwb) from precipitation and potential evapotranspiration. Next, they defined that 15% of monthly cwb is surface runoff. This is weird and calls for explanation! (Also, according to fig6 there is always flow. According to Fig 4, the annual cwb is negative (PET=1459, PREC=1266). That means that there are months with 0mm surface runoff and eventually 0mm total discharge, please explain.). To obtain streamflow, the authors multiply surface runoff with catchment area. Is this an accepted method? Are there references? Please, explain. In addition, the unit surface runoff in eq2 is mm/d. But it is unclear, monthly surface runoff data were converted into daily values. Also, what is the unit of A (basin area)?

*Response: The observed streamflow generation method has been removed, as measured streamflow data from a the nearest gauge station (River Offin) is being used. This so as we employ the spatial proximity setting approach aimed at obtaining a more realistic streamflow data for our analysis. The basin area (A) had the units of m². It should be noted that, the chances of rains are low during the dry season (November to February) in the Kumasi metropolis since the entire area country (Ghana) is dominated by a high pressure system and the North Easterly trade winds. Therefore, surface runoff is also likely to be low and discharge will reduce to a minimum.*

P7,l18: There must be two model sets, for daily and for monthly calibration, right? But later on, I see only one parameter set.

*Response: The results for calibration and validation for both model sets are shown in pages 10-11.*

P10Table4,5: Are these tables really necessary?

*Response: Absolutely, since they clearly show the ranking of the sensitive parameters.*

P11, l3: It is unclear to me, why the authors calibrated against daily and monthly streamflow data, and only monthly results are presented. Due to the uncertainty of "measured" streamflow, I would only calibrate at a monthly scale and delete all passages related to daily calibration.

*Response: The new model run which is still underway would focus only on the monthly simulations of streamflow since observed streamflow data from the River Offin gauging station is available in mean monthly resolution.*

P11, l13: 10 and 12 mm maximum rainfall per month? This is certainly much too low!!!

*Response: The values are mean monthly, which was unfortunately not stated in the text. This has been modified in the revised manuscript.*

P12, fig7: maximum monthly rainfall approx. 18 mm? This is certainly much too low!!!

*Response:    It is the maximum mean monthly rainfall value.*

P11,l20-21: I do not understand, why fitted parameters were inserted into a default model? "This was to ensure that the new fitted ranges boosted the model performance for climate and stream-flow prediction." I cannot follow, why model performance for climate? Please, explain!

*Response: The aim was to observe the trends in the water balance parameters as well as streamflow after model calibration. Climate prediction has been removed.*

P12,l1: I am confused: There is Fig6 with optimized streamflow and there is fig9 with optimized streamflow data. Predicted streamflow in Fig9 is much higher. Why showing uncalibrated data…

*Response:  The model is currently being re-run, and this section would be modified.*

P13,l5: Why did the authors only used RCP8.5 and not others?

*Response: The high pace of urbanisation and deforestation at the catchment led to the choice of the RCP8.5 as the projection scenario. Notwithstanding, we are also including trends in RCPs 2.6 and 4.5 in the revised manuscript. Previously, the landuse input map was used for the projection of future hydro-climatic trends. However, different landuse scenarios are being developed in hand with the different RCPs to give consolidated conclusions to this study.*

Table7: The table captions are not explained. Is it important to show lateral flow? Groundwater flow is not show. Is the high percentage of surface runoff realistic? WYLD in future projection is more then three times larger. Is this realistic? The same for surface runoff….

*Response: Table captions would be explained. Meanwhile, the model is currently being re-run, the results will be modified to reflect the future hydrology of the catchment.*

Figure 10: I believe it is not a good idea to show future projections year by year. Nobody knows, how the weather will be in 2040. But there are scenarios, how the climate will develop. Therefore, I suggest to compare a 30-years reference period (average) with the future projection (also 30 years, 2021-2050)

***Response: This would be done.***

**REFERENCES**

Arnold, J., Kiniry, J., Srinivasan, R., Williams, J., Haney, E., and Neitsch, S.: Soil and Water Assessment Tool: Input/Output Documentation version 2012, Texas Water Resources Institute, 2012.

Aryee, J.N.A., Amekudzi, L.K., Quansah, E., Klutse, N.A.B., Atiah, W.A. and Yorke, C., 2017. Development of high spatial resolution rainfall data for Ghana. *International Journal of Climatology*.